# Processing Hundreds of SARS-CoV-2 Samples with an In-House PCR-Based Method without Robotics

**DOI:** 10.3390/v13091712

**Published:** 2021-08-28

**Authors:** Theresa Mair, Maja Ivankovic, Christian Paar, Helmut J. F. Salzer, Angelika Heissl, Bernd Lamprecht, Elisabeth Schreier-Lechner, Irene Tiemann-Boege

**Affiliations:** 1Institute of Biophysics, Johannes Kepler University, 4020 Linz, Austria; maja.ivankovic@stud.sbg.ac.at (M.I.); Irene.Tiemann@jku.at (I.T.-B.); 2Institute of Laboratory Medicine, Johannes Kepler Universitätsklinikum, 4020 Linz, Austria; Christian.Paar@kepleruniklinikum.at (C.P.); Elisabeth.Schreier-Lechner@kepleruniklinikum.at (E.S.-L.); 3Department of Pulmonology, Johannes Kepler Universitätsklinikum, 4020 Linz, Austria; Helmut.Salzer@kepleruniklinikum.at (H.J.F.S.); Bernd.Lamprecht@kepleruniklinikum.at (B.L.); 4Department of Biosciences, University of Salzburg, 5020 Salzburg, Austria; angelika.heissl@sbg.ac.at

**Keywords:** SARS-CoV-2 PCR detection, COVID-19 pandemic, magnetic bead RNA extraction, RT-qPCR, TaqMan chemistry

## Abstract

The SARS-CoV-2 pandemic has required the development of multiple testing systems to monitor and control the viral infection. Here, we developed a PCR test to screen COVID-19 infections that can process up to ~180 samples per day without the requirement of robotics. For this purpose, we implemented the use of multichannel pipettes and plate magnetics for the RNA extraction step and combined the reverse transcription with the qPCR within one step. We tested the performance of two RT-qPCR kits as well as different sampling buffers and showed that samples taken in NaCl or PBS are stable and compatible with different COVID-19 testing systems. Finally, we designed a new internal control based on the human *RNase P* gene that does not require a DNA digestion step. Our protocol is easy to handle and reaches the sensitivity and accuracy of the standardized diagnostic protocols used in the clinic to detect COVID-19 infections.

## 1. Introduction

In late 2019 and early 2020, the first cases with viral pneumonia of unknown cause were reported in Wuhan, China, which were later identified as being caused by the novel SARS-CoV-2 virus (severe acute respiratory syndrome coronavirus 2) [1,2], and has since caused a severe global pandemic. SARS-CoV-2 is a highly infectious coronavirus that causes the disease COVID-19 (coronavirus disease 2019) with the most common symptoms fever, dry cough, and fatigue, as reported by the WHO. It was shown that this virus is most closely related to SARS-like coronaviruses in bat and civet within the family *Betacoronavirus*, subgenus *Sarbecovirus*, but more distantly related to SARS-CoV and MERS-CoV [3,4], two known highly pathogenic coronaviruses that have been identified in humans in 2003 and 2012, respectively, causing severe respiratory diseases [5,6].

The high reproductive rate of SARS-CoV-2 resulted in a world-wide spread of the virus that is being contained by social isolation, vaccination, and extensive testing. To date, many testing strategies have been developed that use nasal or throat swabs. Those tests are based on the detection of the viral RNA (PCR-based tests) and other cheap, fast virus tests that measure viral antigen or antibodies and return a result within 15 min on a paper strip. These latter tests are a qualitative measure of infection or no-infection and detect only very high viral loads. Moreover, these rapid tests have been calibrated with different PCR systems and the percentage of positives detected with these tests has dropped when a more sensitive PCR system was used as a reference [7]. Thus, scientists debate whether these millions of cheap, fast diagnostic kits will help to control the pandemic, given that false negative results might spread the virus, leading to a false sense of security [8]. The testing strategy also depends on infection rates in their area, with rapid tests implemented to detect and isolate highly infectious people, but rapid tests are not reliable enough for assessing the prevalence of the virus in the population. Thus, PCR-based tests are still considered the golden standard to detect SARS-CoV-2 and will stay as the main test for screening the pandemic [8].

One of the shortcomings of PCR-based tests is the long and laborious processing time. Several research groups in the world are simplifying and shortening the PCR test, such as the approach based on a technique called loop-mediated isothermal amplification, or LAMP, which is faster than PCR and requires minimal equipment. However, these tests are not as sensitive as PCR tests performed with a thermocycler or quantitative PCR (qPCR) [9,10].

The qPCR-based COVID-19 diagnostic tests are often run in closed systems (e.g., Roche COBAS system) requiring expensive equipment and reagents that have faced shortages. Moreover, most available protocols for high throughput testing require robotic components that are expensive or not available in standard laboratories. Here, we report a qPCR-based method for detecting SARS-CoV-2 viral RNA in naso-pharyngeal swabs using an open system compatible with different alternative components and reagents. Moreover, we implemented the use of multichannel pipetting devices in combination with magnetics for the rapid sample processing that also reduces handling and potential sources of error and sample mix-up. Our protocol is partially based on protocols published by the Charité [11,12] and the US Centers for Disease Control and Prevention (CDC) 2019-nCoV Real-Time RT-PCR Diagnostic Panel Instructions for Emergency Use Authorization. In addition, we developed a new set of probes for the internal positive control that span two exon boundaries of the human *RNase P* gene, eliminating the need of a DNA digestion step. Our protocol was extensively validated with patient samples run in parallel in the clinic (Kepler Universitätsklinikum, KUK) on the COBAS system (Roche, Basel, CHE) and has a very high reliability (no false positives) and sensitivity. In conclusion, our validated protocol reaches similar levels of performance of the COBAS SARS-CoV-2 diagnostic tests, but uses a simple hands-on procedure with a capacity of processing of hundreds of samples per day in a lab with standard quantitative PCR equipment and no robotics.

## 2. Materials and Methods

### 2.1. Test Material and Lysis Buffers

Naso-pharyngeal swabs from patients in different buffers (COBAS, NaCl, PBS, and Copan) were collected. Different lysis buffers and media have been tested during this study. Buffers and their compositions are listed in Table 1.

### 2.2. Primers and Probes

Different primer/probe sets suggested by the CDC and Corman et al. [12] have been used targeting regions within the *N-gene* (N1 and N2) or *E-gene* (E-Sarbeco) of SARS-CoV-2 and two regions within the human *RNase P* gene as an internal control (RP) with one of them (RP_2) that has been designed by us considering exon–exon boundaries (Table 2). For N1, N2, and RP, premixed primer/probe sets were ordered at Integrated DNA Technologies (IDT, Coralville, IA, USA, 2019-nCoV CDC EUA Kit, #225521231), whereas primers and probes for RP_2 and E-Sarbeco have been ordered individually at IDT and mixed in proper ratios afterwards.

### 2.3. Viral RNA Extraction

For the viral RNA extraction, the Sera-Xtracta Virus/Pathogen Kit (Cytiva, Marlborough, MA, USA) was used. Note that a 96-well plate format was used for processing more than 12 samples including the use of electronic multichannel pipettes. For a lower sample number, single 1.6 mL low-binding tubes can be used. First, SeraSil-Mag 400 and SeraSil-Mag 700 beads were mixed in a 1:1 ratio and carefully vortexed before adding to any sample. Measures of 100–400 µL sample in collection buffer were then mixed with 20 µL of bead stock mixture and 570 µL of Binding/Lysis Reagent in a 96-well deep-bottom plate (Thermo Fisher, Waltham, MA, USA, #95040450; GE Healthcare, Chicago, IL, USA, #7701-5200). In order to enhance cell lysis, 10 µL of 20 mg/mL Proteinase K Solution can be added. The samples were mixed carefully by pipetting up and down several times. Only if the Proteinase K Solution was added, the plate was sealed with a clear adhesive film (Thermo Fisher, #4306311; or GE Healthcare, #7704-0001) and the samples were incubated at 70 °C for 3 min using a water bath followed by centrifugation for 2 min at 2000× *g*. In the next step, the plate was placed on a Magnet Stand-96 (Thermo Fisher, #AM10027) for at least 2 min (or until the solution became clear). In case of a sealed plate, the foil is pierceable and does not need to be removed. Without disturbing bound beads, the supernatant was carefully removed and discarded. Then, the plate was removed from the magnet and 950 µL Wash Buffer was added and mixed with the beads carefully by pipetting up and down. The plate was put back on the magnet for at least 2 min and the supernatant was discarded. In order to wash the beads again, the plate was removed from the magnet and 950 µL freshly prepared 80% ethanol was added and mixed with the beads carefully. After placing the plate back on the magnet for another 2 min, the ethanol was discarded and the beads were dried for 2 min, staying on the magnet. In this step, it is crucial that all the ethanol is completely gone. If necessary, the remaining ethanol droplets were removed using small tips. To finally elute the pure RNA, the plate was removed from the magnet and 50 µL of nuclease-free water (Sigma-Aldrich, St. Louis, MO, USA) was used to carefully resuspend all the beads. The plate was placed on the magnet one last time for 2 min and 48 µL eluate were transferred to a fresh 96-well Elution plate U-bottom (Macherey-Nagel by GE Healthcare, Düren, Germany, #740486.24). The Elution plate was sealed for proper storage. Note, when working with a 96-well format and multichannel pipettes, nuclease-free pipetting reservoirs were used (Channel Mate by StarLab, Hamburg, Germany, #E1306-2510).

### 2.4. One Step RT-qPCR

For RT-qPCR, the Luna Probe One-Step Reaction Mix from New England Biolabs (NEB, Ipswich, MA, USA, #E3007E) or the TaqMan Fast Virus 1-Step Master Mix from Thermo Fisher were used. In a 10 µL reaction, either 1xLuna Probe One-Step Reaction Mix and 1xLuna WarmStart^®^ RT Enzyme Mix or 1xTaqMan Fast virus 1-step master mix were supplemented with 0.5 µM of forward and reverse primer and 0.13 µM probe using the respective primer/probe mix for each target. As a template, 2.5 µL extracted RNA was added. In each experiment, separate reactions for at least three targets in total were prepared: two virus-specific targets (N1, N2 or E-Sarbeco) and one control target (RP or RP_2; for primer/probe information, see Table 2). In multiplex reactions, the primer/probe concentration was 0.5 µM and 0.13 µM per each primer and probe.

As a negative control, one reaction with nuclease-free water instead of RNA was performed for each target (NTC, no template control); as a positive RNA control, 10^5^, 10^4^, 10^3^, and 10^2^ copies of Twist Synthetic SARS-CoV-2 RNA Control 1 (Twist Bioscience, South San Francisco, CA, USA, #MT007544.1) were used. The RT-qPCR reactions were set up in an optical 384-well plate (Bio-Rad, Hercules, CA, USA, #HSP3865) on ice and sealed with adhesive optical clear qPCR plate seals (Bio-Rad, #MSB1001), briefly mixed and centrifuged for 2 min at 2000× *g*. The RT-qPCR run was performed using a CFX384 Real-Time System type C1000 thermocycler (Bio-Rad) with the following thermal conditions: 25 °C for 2 min, 50 °C for 20 min, 95 °C for 3 min and 45 cycles of 95 °C for 15 s and 58 °C for 10 s including a plate read to detect for the FAM/Cy5/HEX signal, depending on the probes used.

In order to show that our designed RP_2 primer/probe set does not amplify genomic DNA, we also set up real-time qPCR reactions without a reverse transcriptase. Therefore, the Luna^®^ Universal Probe qPCR Master Mix (NEB, #M3004G) was used. The 10 µL reactions consisted of 1x Luna Universal Probe qPCR Master Mix, 0.5 µM of forward and reverse primer and 0.13 µM probe filled up with nuclease-free water. All further steps were identical and as already described.

### 2.5. CFX Plate Loading Strategy

In order to run a full plate, a maximum of 88 samples can be processed together with four dilutions of the RNA control (10^5^, 10^4^, 10^3^, and 10^2^ copies) and one NTC for each primer/probe set. For optimized sample handling and transfer from the 96-well Elution plate after RNA extraction to the 384-well qPCR plate using multichannel pipettes, we developed a pipetting scheme shown in Figure 1. We avoided using border wells of the qPCR plate as fluorescence signals are not perfectly detected there.

### 2.6. Data Analysis

The CFX Manger 3.1 Software (Bio-Rad) was used to monitor the data. Results were then analysed according to the guidelines of the CDC that have been approved by the US federal Drug administration for emergency use as follows: where ‘+’ means that the amplification with the corresponding primer/probe set was detected with Cq < 45 and ‘-‘ means that no amplification was detected (see Table 3).

### 2.7. Room Separation and Decontamination

In order to avoid contaminations of any kind, it is crucial that protocol steps follow a certain workflow including the usage of separate and specifically assigned lab areas. For this reason, sample processing, RNA extraction, and RT-qPCR were performed in different rooms equipped with an MSC-Advantage Class II Biological Safety Cabinet, 1.8 m. Samples that had not been treated for an inactivated/lysed virus were processed in a S2 safety hood first. Before and after usage, the biosafety laminar flow hoods were cleaned with DNA-off (Takara, Kusatsu, Japan) and sterilized under UV light for 10 min. This procedure ensures the destruction of any potential contaminating RNA or DNA. Note that the fast, innate degradation of RNA significantly reduces the potential of contamination from this molecule. In laminar flow hood 1 (safety hood 1) the RT-qPCR master mix was prepared that was then taken into laminar flow hood 2 (safety hood 2). Patient samples stored in buffers/transport media still including active virus particles were taken to the S2 safety hood (safety hood 3) and the first steps of RNA extraction were performed. RNA extraction was then finished in safety hood 2 and pure extracted viral RNA was added to the RT-qPCR reactions aliquoted to the 384-well plate, which was finally processed in the CFX thermocycler (Figure 2). Plates were sealed properly when they were taken out of hoods and were not allowed to be opened after PCR amplification was finished. To prevent sample contamination (1), each room is used exclusively for the application or technique indicated and (2) the workflow is always unidirectional (indicated by the arrow). Additionally, equipment and consumables are not interchanged between the different laboratory rooms and spaces, and accordingly, each room has its own pipettes, tip boxes, gloves, and consumables.

## 3. Results

### 3.1. Principle of Workflow

The principle of our in-house open-system PCR-based assay to detect SARS-CoV-2 viral RNA was adapted from the process of the Drosten group [11,12] and CDC. In order to validate this assay, our gold standard was the COBAS 6800 system from Roche (COBAS system hereafter), which was used for SARS-CoV-2 diagnosis in the Institute of Laboratory Medicine, Kepler Universitätsklinikum (KUK). The procedure uses a naso-pharyngeal swab taken from patients, which was added to either a lysis buffer that inactivated the virus (guanidinium based) or to a transport medium (saline solution or PBS) that kept the viral particles intact until further processing. The viral RNA was extracted and purified using magnetic beads in a 96-well format together with multichannel pipettes and 96-well plate magnets. Finally, extracted RNA was transcribed into cDNA by a reverse transcriptase (RT) followed by amplification of specific regions with quantitative real-time PCR (qPCR) in a one-step reaction (RT-qPCR) using primer/probe sets (TaqMan-based chemistry) that specifically recognizes the viral genome. The RT-qPCR was performed in a 384-well format using an in-house qPCR thermocycler, but the reaction can also be performed in a 96-well qPCR thermocycler. For each sample, we performed three RT-qPCR reactions in parallel using two viral specific probes targeting either the *N-gene* (N1, N2) or the *E-gene* (E-Sarbeco) of SARS-CoV-2 and one probe targeting the human *RNase P* gene as an internal positive control (RP or RP_2). The combination of results obtained by the three primer/probe sets was used to categorize a sample as positive or negative for a COVID-19 infection: a sample was called positive if both virus-specific probes were detected; the sample was negative if none of the virus-specific probes were detected; if all three probes (including the internal control) were not detected, the test was categorized as invalid and all other combinations were recognized as inconclusive (Table 3).

With our set-up, we can process one plate per experiment with up to 88 samples in addition to several positive and negative controls. For each reaction, as a positive RNA control, we used a 1:10 dilution series starting with 10^5^ to 10^2^ copies of Twist Synthetic SARS-CoV-2 RNA Control 1, and as negative control, one reaction without any RNA template for each target (two viral targets and one internal control). The combination of plate extractions with multichannel pipetting rendered a high sample throughput, and reduced sample handling and error sources. All details of the experimental procedure are shown in the section Materials and Methods. 

### 3.2. Two Different RT-qPCR Kits Can Be Used

In a first step, we tested the performance of two different one-step RT-qPCR kits: TaqMan™ Fast Virus 1-Step Master Mix from Thermo Fisher (TaqMan kit, hereafter) and Luna^®^ Probe One-Step RT-qPCR Kit from NEB (Luna kit, hereafter). For this purpose, we used a total of 175 samples collected in COBAS buffer, NaCl, or PBS that were measured either with the TaqMan or Luna kit. To assess the performance of both kits, we compared the average Cq (quantification cycle) values resulting from the N1 and N2 virus-specific targets that have been grouped based on the initial Cq of ≤30 (high viral load), >30 (low viral load), or negative (not detected) as it was reported by the COBAS system (Table 4). We could not observe any major differences between the TaqMan and Luna kit. Both rendered similar average Cq values compared to the COBAS system for samples with high viral load: Cq = 23.8, 24.7 and 25.3 (N1, N2 and COBAS, respectively) for the Luna kit and Cq = 28.8, 29.2 and 26.9 (N1, N2 and COBAS, respectively) for the TaqMan kit (Table 4). Additionally, for the samples with low viral loads, the average Cq values of N1 and N2 were similar to the initial Cq values (COBAS) for both kits. Given our results, we recommend the use of either one or the other RT-qPCR kit for the process; although, the Luna kit is more affordable.

### 3.3. Assay Sensitivity: Comparison of Performance of Open System Assay with COBAS 6800

In order to evaluate the robustness of our testing method, we compared in total 219 naso-pharyngeal swabs (stored either in COBAS buffer or NaCl) measured by the KUK with the Roche COBAS diagnostic test (COBAS) or with our in-house assay (JKU); data shown in Appendix A. For samples with a high viral load (Cq ≤ 30), the sensitivity of our in-house test was 100% and all 53 samples were identified by both platforms as positive (Table 5). Moreover, the obtained Cq values for these samples were very similar with an average Cq of 25.4, 25.1, and 25.8 measured with the COBAS system, the JKU for N1, and the JKU for N2, respectively. Samples with a low viral load (Cq > 30) measured by the COBAS system or the JKU or both were more problematic. From Figure 3A, it can be observed that samples with a Cq > 30 are missed sometimes in one or the other platform; however, a positive result is missed more often by our in-house than by the COBAS platform. Of the 71 samples with a Cq > 30, the JKU platform identified fewer samples (46% less) as positive compared to the COBAS platform and samples were instead categorized either as “inconclusive” (only one of the two PCR assays rendered an amplification signal) or “negative”. In particular, samples with a Cq > 35 showed the most inconsistency between measurements. Interestingly, some of these samples were inconsistent also between repeated measurements with our system (Appendix A).

Finally, we also evaluated the specificity of our testing system via a ring trial, which uses material of different viral strains. These are used routinely in diagnostic tests for quality control. We participated in such a ring trial (INSTAND EQA scheme (340) Virus Genome Detection—Coronaviruses incl. SARS-CoV-2—June/July 2020) with our in-house assay with the samples provided by the KUK. The same samples were also measured with the COBAS diagnostic test system (Figure 3B and Appendix A). The results of the ring trial are concordant with the expected outcome and also agree with the results of the COBAS system, reflecting the specificity of our approach.

### 3.4. Technical Sensitivity and Reproducibility of Our Open System Assay

In order to test the technical sensitivity and reproducibility of the JKU test, we first prepared a dilution series containing 10^5^, 10^4^, 10^3^, and 10^2^ copies of synthetic SARS-CoV-2 RNA control and carried out the RT-qPCR step in four or eight replicates of these controls (Figure 4A). Our results showed that our system can measure as low as 100 RNA molecules for both virus-specific probes N1 (blue) and N2 (orange) in the RT-qPCR step (Appendix A). We also observed a constant amplification efficiency between samples with an estimated efficiency of E = 88–93%, estimated as E = 10^[−1/slope]^ − 1. The very high correlation (R^2^) above 0.99 between replicates and dilutions also indicates a high reproducibility at the level of reverse transcription and PCR. The standard deviation measured between four and eight control replicates ranged between 0.15 and 0.47 Cqs (Appendix A). Similar results were obtained for a dilution series of extracted RNA of two positive patient samples (collected in COBAS buffer) with an initial Cq of ~21–26 diluted by three orders of magnitude run in duplicates. Both N1 and N2 probes rendered similar average Cq values (Appendix A). However, in the range of 37 Cq or more, one or both reactions for N1 and N2 did not amplify (inconclusive result).

To further explore the reproducibility and repeatability of our assay, we performed in total 98 extraction replicates using pre-screened patient samples collected in COBAS buffer, NaCl, or PBS. Of those, 67 samples were positive (18 with high viral loads, Cq ≤ 30; 49 with low viral loads, Cq > 30) and 31 samples negative. We had a good agreement between replicates, rendering very similar average Cq values for the two virus-specific probes N1 and N2 for samples with high viral loads (Table 6 and Appendix A). For samples with low viral loads (Cq > 30), we measured a positive signal for either N1, N2, or both for only 50% of the samples, agreeing with our previous observation that at low viral load, the number of false negatives increases.

### 3.5. RNA Extraction Efficiency and Limit of Detection (LOD)

In order to test the RNA extraction efficiency of our assay, we first prepared a pool of negative patient samples collected in COBAS lysis buffer and added different dilutions of synthetic SARS-CoV-2 RNA before the RNA extraction step. The samples contained 10^6^, 10^5^, 10^4^, 10^3^, and 10^2^ copies of RNA per reaction in the same RNA/DNA negative patient sample pool. We observed a high linear correlation (R^2^) of 0.998 between the input RNA and the respective Cq value for both N1 and N2 (Figure 4B and Appendix A); however, for N1, we could not detect the lowest RNA concentration of 10^2^ RNA molecules/reaction, defining the limit of detection (LOD) of our assay at 10^3^ to 10^2^ copies of RNA. Interestingly, when converting the Cq values to RNA input amounts with the standard curve (10-fold dilution series starting with 10^5^–10^2^ RNA molecules), we estimated that RNA molecules are lost by ~100 fold. In other words, the amplification between the same RNA input with and without extraction step is delayed by around five cycles (Figure 4B).

Next, we tested if our system is less sensitive to samples with lower viral load. For this purpose, we prepared before extracting the viral RNA a 1:5 to 1:625 dilution series in fivefold steps in buffer 6 of two positive patient samples with an approximate viral load of 10^5^ to 10^6^ (Cq ~26 and ~25) (Figure 4C and Appendix A). The Cq was reduced proportionally with our dilution steps, and we observed a high linear correlation factor R^2^ of ~0.99 for all primer/probe sets for the two samples. The results indicate that our assay can measure samples with low viral loads (10^3^–10^4^).

### 3.6. Lysis Buffer Can Affect the Test Performance

The buffer used to store naso-pharyngeal swabs is an important aspect of the SARS-CoV-2 testing pipeline. When processing RNA, the chemical guanidinium thiocyanate (GITC) or guanidine-HCl is an important component to destroy RNases; however, these components are expensive and toxic. Alternatives based on detergents (e.g., Triton) or stabilizing solutions (PBS or NaCl) have been proposed. Thus, here, we tested different lysis buffers for sample collection compatible with the Roche COBAS system, as well as with our in-house system (JKU). For this purpose, we used a pool of positive samples (n = 12; average Cq 19) added in different dilutions (1:10^2^, 1:10^3^, 1:10^4^, 1:10^5^) to negative samples stored in ten different buffers (for buffer compositions, see Table 1). We tested stabilizing buffers in saline solution or transport media (e.g., NaCl and VTM, viral transport medium), as well as lysis buffers with detergents and/or GITC or guanidine-HCl. We measured the diluted samples with the COBAS system at the clinic and with our in-house assay. Eight out of ten buffers rendered very similar results, with the COBAS buffer having the highest sensitivity, rendering slightly lower Cqs with the most diluted samples (Figure 5). In contrast, the highest dilution (1:10^5^) was not detectable in samples stored in buffer 2 and VTM. Our experiments suggest that the lysis buffer might influence the RNA extraction efficiency and detection sensitivity. In this work, we also show that different buffers are compatible with the SARS-CoV-2 detection and can be used interchangeably without affecting the assay performance.

### 3.7. Sample Stability

The coronavirus pandemic came along with a recurring increase in infected COVID-19 patients preceding an even higher number of people potentially positive for SARS-CoV-2. In many cases, the large sample numbers created a bottleneck and samples had to be stored for a few days. Thus, part of our study was to assess the stability of samples stored over several days. For this purpose, we tested a total of 35 samples collected in PBS buffer at day 1 and day 3 stored at 4–8 °C. Of these, 15 were reported to be negative and 20 to be positive by the COBAS system (12 with a high viral load, Cq ≤ 30; 8 with a low viral load, Cq > 30; Table 7 and Appendix A). For our measurements, we used the N2 and E-Sarbeco virus-specific probes. Out of the 15 negative samples, we detected the same samples to be negative at day 1 and only one sample to be positive at day 3, however, with a high Cq value of ~37, which represents samples with low viral load often missed by multiple testing. From 12 positive samples with high viral loads (average Cq 24), we detected the same samples to be positive at day 1 and day 3 with average Cq values of 25.8 and 26.9, respectively. For low viral load samples, we identified three out of eight samples (Cq 33.2) as positive at day 1 and 3 with Cq values of 32.7 and 33.8, respectively. The remaining five samples were identified either as negative or gave inconclusive results on both days. Our results showed that the average Cq values agreed with the values reported by the COBAS system measured at the same day and also three days later, suggesting that collected patient samples are stable for at least three days in PBS buffer when storing at cooling conditions. Moreover, samples are not only stable in PBS buffer, but also in other buffers that have been used in this study, such as COBAS or NaCl (data not shown).

### 3.8. Primer Test

The CDC recommended that the primer/probe set for the human *RNase P* gene, which serves as an internal positive control for RNA extraction and reverse transcription (RT), is suboptimal as both primer binding sites are located within exon 1. Thus, this design requires the use of a DNase step to eliminate any signal coming from human DNA. However, this introduces an extra step and source of error. If a DNA digestion step is not used, the amplification will mainly come from the co-extracted genomic DNA instead of the human RNA, resulting in a false-positive signal for the internal control.

We designed a new primer/probe set (RP_2) specific for the human *RNase P* mRNA with primers placed on different exons. The reverse primer spans an exon–exon junction, solely targeting the mRNA sequence (Figure 6A). To prove the specificity of our primer/probe set, we set up real-time qPCR reactions with and without a reverse transcriptase using naso-pharyngeal swabs in PBS as samples (15 negative and five positive patients provided by the KUK). As seen from Figure 6B, the RP CDC primer/probe set signal was largely independent of the reverse transcription step (average Cq with and without RT: 29.25 ± 2.13 and 29.14 ± 2.17, respectively), indicating that most of the signal results from amplification of genomic DNA. Note that here, we did not include a DNA digestion step. As expected, we did not observe an amplification with the RP_2 primer when the reverse transcriptase (RT) step was omitted (n = 20). When we included the RT step, we detected a signal for 10 out of 20 samples (average Cq: 36.0 ± 1.36), showing the specificity for the human *RNase P* cDNA sequence (Appendix A). With our RP_2 primer/probe set, six patients were classified as invalid as no *RNase P* control could be detected and the sampling was likely not properly performed.

In order to increase throughput, we attempted multiplex reactions by attaching different fluorophores to the probes. We were unsuccessful combining N1 (FAM), N2 (HEX), and RP_2 (Cy-5) into a single reaction and there was a crosstalk between reactions, resulting in the amplification of artefacts since negative samples and the non-template control resulted in an amplification signal for N1 and N2 (Cq ranging from 33 to 38). The duplex reactions N1+RP_2 and N2+RP_2 were unsuccessful for the same reasons (data not shown).

## 4. Discussion

### 4.1. Sample Handling

One of the major concerns when using manual protocols in a clinical lab setting is the sample handling, in which sample mix-up, pipetting errors, or other human mistakes could jeopardize the method. To minimize these sources of human errors, robotics have often replaced the human labour. With our method, we introduced the use of electronic multichannels, plates, and magnetics. Note that a plate format is used for processing more than ~12 samples. For fewer samples, we used 1.6 mL low-binding tubes. Throughout, the protocol samples were handled with 8- or 12-channel electronic multichannel pipettes.

Multichannel pipetting within plates has the advantage over single-channel pipetting that several samples can be processed simultaneously. This simultaneous handling reduces sample skipping or the use of the wrong wells. Furthermore, fewer pipetting steps are necessary, increasing the speed and time of sample processing, which also reduces extensive pipetting that leads to human fatigue. In addition, in our protocol, we implemented the use of electronic multichannel pipettes as described in the Standard Operating Procedure (SOP) in Appendix A. The features of these pipettes such as pipetting speed and sample release is controlled and standardized between samples and experiments. Moreover, simple pipetting programs (e.g., used for sample mixing) allow for a semi-automated processing similar to the one implemented by robotics. This is of particular importance especially for the RNA-extraction protocol that requires several pipetting steps and could be an important source of human error and pipetting fatigue. With our semi-automated process, these risks were minimized. Overall, our protocol takes 1–2 h of handling time, depending on the experience and skills of the technician without the RT-qPCR thermocycler run time. This is comparable to the time used by the COBAS robotics system (internal communication) and other robotic systems; however, the cost of electronic multichannel pipettes is a fraction of pipetting robots.

Another important step in sample processing is data capture. To also minimize sample mix-up and human error, each tube with the naso-pharyngeal swab had a barcode. The barcode of each swab was scanned, and the sample was placed in a rack. The position in the rack was linked to the barcode and samples were further processed in 96-well and 384-well plate formats, such that a sample always stayed at a defined position throughout the whole process. The barcodes were recorded in an electronic tabular file, which was imported into the CFX real-time PCR program and used for the data output, which linked a Cq value with the specific barcode of a sample. This improved workflow increased productivity, and reduced costs and incorrect sample identification.

### 4.2. Reproducibility at Low Viral Loads

Our results showed that our system can measure as low as 100 RNA molecules of the virus. Note that this limit is at the RT-qPCR step. In other words, as low as 100 freshly extracted viral RNA copies can be reverse-transcribed and amplified in PCR, rendering an amplification product before 40 cycles. However, we noticed that at the extraction step (lysis of the viral particle and clean-up of the viral RNA), a large number of viral particles were lost, either by inefficient lysis or binding to the magnetic beads. Based on our experiments testing the extraction efficiency with synthetic viral RNA measured with and without the extraction step, we observed a loss of sensitivity by at least 100-fold, equivalent to ~5 Cqs. This drop of sensitivity due to the loss of viral copies during the extraction steps also explains the lower sensitivity of patient samples with low viral loads (Cq ~30–35) with inconsistencies between measurements becoming a re-occurring problem especially for samples with Cq > 35. This also led to a lower reproducibility between replicates or different targets (N1 and N2) in samples with low viral loads. For high viral loads (≤ 30), both the COBAS and our in-house system rendered reproducible Cq measurements with a 100% agreement between platforms, experiments, and targets (N1 and N2), including replicates performed a few days apart. The loss of reproducibility is a common denominator for any testing system reaching its LOD. Similar to our in-house method, many PCR-based testing protocols have an LOD at viral loads of ~10^3^ [8,12]. Interestingly, PCR-based tests that do not included a lysis/RNA extraction step and directly add a sample of the patient swab to a RT-qPCR reaction are less sensitive than protocols including an RNA extraction step [9,10].

## 5. Conclusions

We developed a PCR test to screen COVID-19 infections that does not require expensive robotics and can still process hundreds of samples per day with the use of multichannel pipettes and plate magnetics for the RNA extraction step and RT-qPCR step. We also tested two different kits that combine the reverse transcription step with the qPCR. Furthermore, we also evaluated different sampling buffers and showed that samples in NaCl or PBS are stable for at least 3 days. Finally, our design of the internal positive control does not require a DNA digestion step and faithfully represents the RNA extraction and detection efficiency. Our protocol is easy to handle and reaches the sensitivity and accuracy of the standardized diagnostic protocols used in the clinic to detect COVID-19 infections.

## Figures and Tables

**Figure 1 viruses-13-01712-f001:**
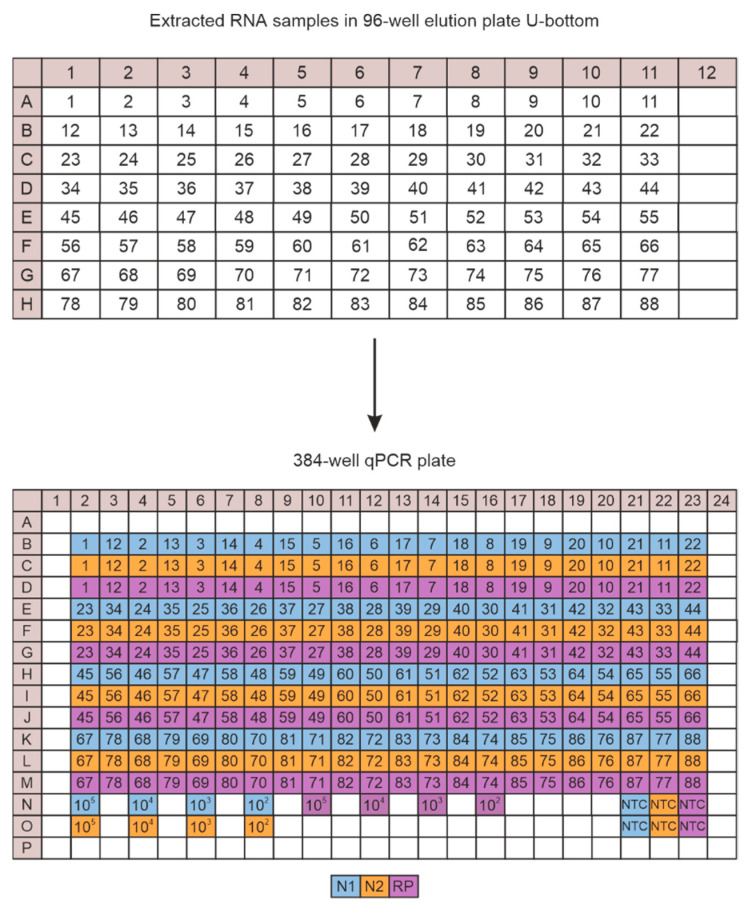
CFX plate loading scheme. Shown is the pipetting strategy for the transfer of extracted RNA from the 96-well elution plate to the 384-well qPCR plate using a 12-channel pipette in order to run a full plate RT-qPCR experiment that includes 88 samples measured with three primer/probe sets like N1 (blue), N2 (orange) and RP (purple). Four dilutions (10^5^, 10^4^, 10^3^, and 10^2^ copies) of the RNA control as well as two no-template control reactions (NTC) for each primer/probe set are included.

**Figure 2 viruses-13-01712-f002:**
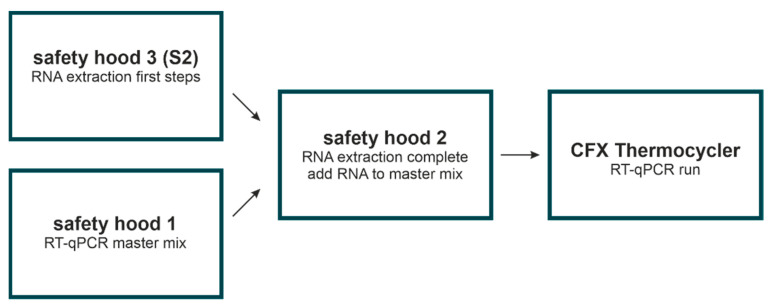
Room separation. Shown are the different steps of our in-house protocol performed in specific rooms to prevent sample contamination and ensure sample and personnel safety.

**Figure 3 viruses-13-01712-f003:**
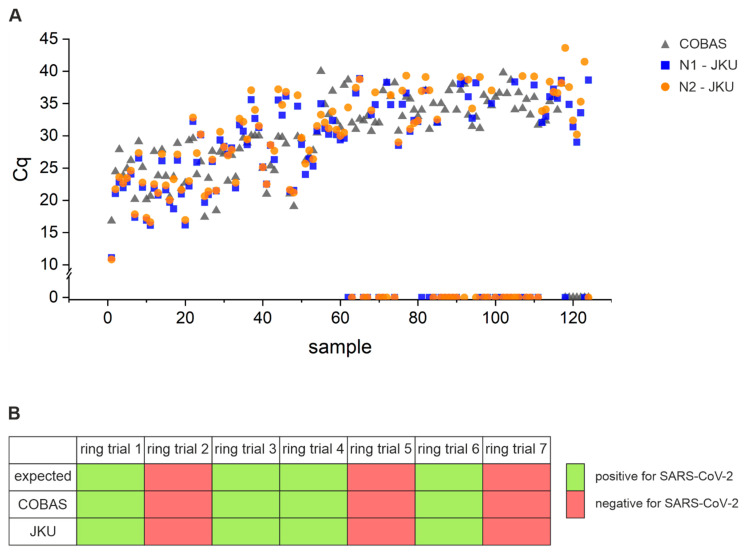
(**A**) 124 SARS-CoV-2 patient samples with an amplification measured either by the KUK with the Roche COBAS diagnostic test (COBAS) and/or our in-house platform for two different targets (N1-JKU and N2-JKU) are displayed with their respective Cq value. Marks on the *x*-axis represent samples without an amplification and are likely false negatives. (**B**) A ring trial was performed with seven samples containing different viral strains provided by the KUK. Samples were classified as positive (green) or negative (red) for SARS-CoV-2, and compared to the expected outcome, the results reported by the diagnostic COBAS system and our protocol (JKU). Note that we measured these samples in duplicates using the two different RT-qPCR kits (Luna and TaqMan) for each replicate.

**Figure 4 viruses-13-01712-f004:**
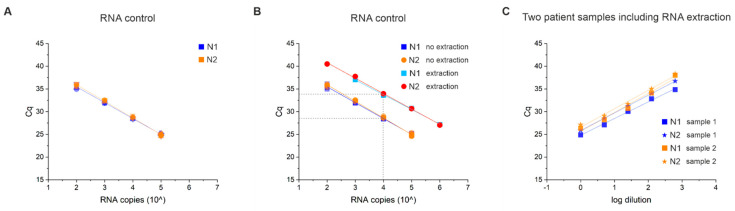
Technical sensitivity, reproducibility, and efficiency of our assay. (**A**) SARS-CoV-2 RNA control was diluted in several fold dilution steps (10^5^, 10^4^, 10^3^, and 10^2^ copies) and measured with RT-qPCR in four or eight replicates (different symbols characterize the different replicates) using the virus specific N1 (blue) and N2 (orange) primer/probe sets. Cq values were plotted against viral synthetic RNA copies. (**B**) SARS-CoV-2 RNA control was added to a pool of negative patient samples to have 10^6^, 10^5^, 10^4^, 10^3^, and 10^2^ RNA copies/reaction before the RNA extraction step followed by RT-qPCR. Cq values for N1 (light blue, including RNA extraction) and N2 (red, including RNA extraction) were monitored and plotted against the increasing number of RNA copies used. The Cq values are compared with the measurements of RNA copies that have been measured without including an RNA extraction step (see plot (**A**); N1, blue; N2, orange). The dotted line visualizes the difference in Cq values of detecting the same number of RNA molecules (e.g., 10^4^ copies) that are either present before (N1 and N2, no extraction) or after (N1 and N2, extraction) RNA extraction. (**C**) Two positive patient samples collected in COBAS lysis buffer were diluted in a series of 1:5 (1:1, 1:5, 1:25, 1:125, 1:625) using buffer 6 before extracting viral RNA followed by RT-qPCR. Detected Cq values for N1 (square) and N2 (asterisk) of both samples (sample 1, blue; sample 2, orange) were plotted against the dilution factor in a log scale. For all plots, the linear correlation (R^2^) was higher than 0.98.

**Figure 5 viruses-13-01712-f005:**
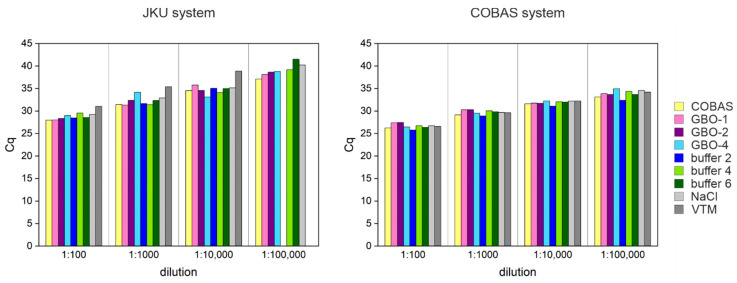
Effect of lysis buffer on test performance. Shown are measured Cq values of a positive sample pool diluted in different lysis buffers in a 1:10 dilution series. Different colored bars in the plot represent the buffers used measured with our in-house system (JKU system) and the Roche COBAS system.

**Figure 6 viruses-13-01712-f006:**
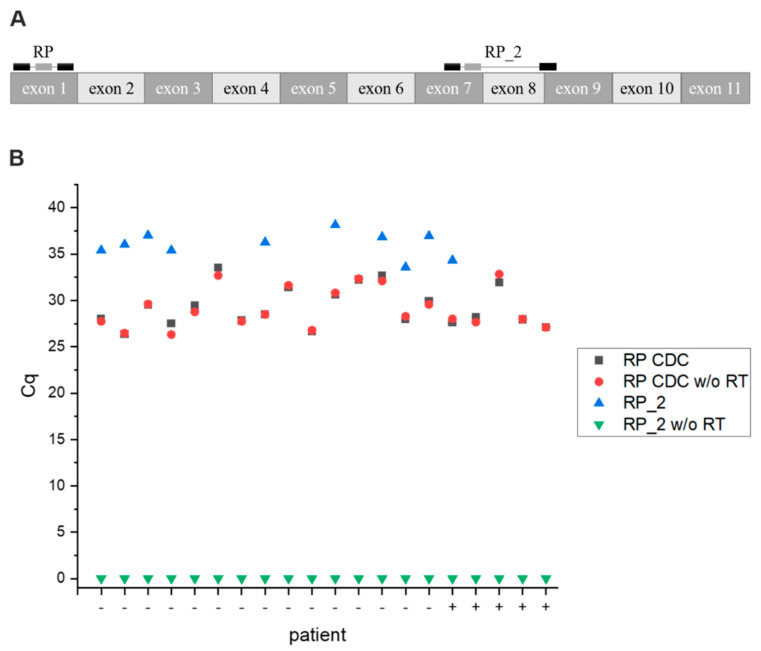
(**A**) For primer design, the sequence of the human mRNA of *RNase P*; *RPP30* (NM_006413.5) was used. Primers and TaqMan probes are indicated as black and grey boxes, respectively. (**B**) The RP_2 and the RP primer/probe set were evaluated with and without a reverse transcriptase step in 20 samples (15 negative and 5 positive), showing the specificity of the RP_2 primer/probe set for mRNA.

**Table 1 viruses-13-01712-t001:** Chemical composition of different buffers used in this study.

Buffer	Components	Source
COBAS	≤40% (*w*/*w*) Guanidine hydrochloride	Roche
	Tris-HCl buffer	
GBO-1	Guanidinium-based	Greiner Bio-One, Kremsmünster, AUT
GBO-2	Guanidinium-based	
GBO-4	PBS-based	
buffer 2	5.4 M GITC	Protocol by Vienna BioCenter
	56.8 mM Tris-HCl, pH 6.4	
	33.1 mM EDTA, pH 7.99	
	1.9% Triton X-100	
	54 mM DTT	
buffer 4	0.15 M NaCl	Protocol by Jena (https://www.schubert-group.uni-jena.de/iomc2media/news-seite/corona+2020/preparation+of+buffers+for+viral+rna+extraction+for+detection+of+a+sars-cov-2-infection_schubert_jena+(1).pdf, accessed on 5 July 2021)
	0.01 M Tris-HCl, pH 7.4	
	0.25% Triton X-100	
buffer 6	4.5 M guanidine-hydrochloride NaCl	
	50 mM Tris-HCl, pH7.4	
NaCl	physiological saline solution (0.9% NaCl)	
PBS	phosphate-buffered saline solution	
VTM (viral transport medium)	2% FBS	Falko Schüllner-Apotheke des A.ö. Landeskrankenhauses—Univ.-Kliniken Innsbruck based on the following protocol (https://www.cdc.gov/coronavirus/2019-ncov/downloads/Viral-Transport-Medium.pdf, accessed on 5 July 2021)
	100 µg/mL gentamicin	
	0.5 µg/mL amphotericin B	
	in 1X HBSS with Ca^2+^ and Mg^2+^, no phenol red	
Copan	Amies medium	

**Table 2 viruses-13-01712-t002:** Primer/probe sequences used in this study.

Name	Target	Sequence	Purpose
2019-nCoV_N1-F	N-gene	GACCCCAAAATCAGCGAAAT	N1 forward
2019-nCoV_N1-R	N-gene	TCTGGTTACTGCCAGTTGAATCTG	N1 reverse
2019-nCoV_N1-P	N-gene	FAM-ACCCCGCATTACGTTTGGTGGACC-BHQ1	N1 probe
2019-nCoV_N2-F	N-gene	TTACAAACATTGGCCGCAAA	N2 forward
2019-nCoV_N2-R	N-gene	GCGCGACATTCCGAAGAA	N2 reverse
2019-nCoV_N2-P	N-gene	FAM-ACAATTTGCCCCCAGCGCTTCAG-BHQ1	N2 probe
RP-F	RNase P	AGATTTGGACCTGCGAGCG	RP forward
RP-R	RNase P	GAGCGGCTGTCTCCACAAGT	RP reverse
RP-P	RNase P	FAM–TTCTGACCTGAAGGCTCTGCGCG–BHQ1	RP probe
RP_2-F	RNase P	GCCCTGCTATCAAAGACTCC	RP_2 forward
RP_2-R	RNase P	TGGCCCTCTTATTTCTAAAGGC	RP_2 reverse
RP_2-P	RNase P	Cy5-TCCAGTGCCCTCAATTTGATGCAA-3BHQ1	RP_2 probe
E-Sarbeco-F	E-gene	ACAGGTACGTTAATAGTTAATAGCGT	E-Sarbeco forward
E-Sarbeco-R	E-gene	ATATTGCAGCAGTACGCACACA	E-Sarbeco reverse
E-Sarbeco-P	E-gene	HEX-ACACTAGCCATCCTTACTGCGCTTCG-3BHQ1	E-Sarbeco probe

**Table 3 viruses-13-01712-t003:** Data analysis and interpretation.

Virus-Specific 1	Virus-Specific 2	RP/RP_2	Result
+	+	+	positive
+	+	-	positive
+	-	+	inconclusive
-	+	+	inconclusive
+	-	-	inconclusive
-	+	-	inconclusive
-	-	+	negative/not detected
-	-	-	invalid

Results were analysed according to the guidelines of the CDC, where ‘+’ means that the amplification with the corresponding primer/probe set was detected with Cq < 45 and ‘-’ means that no amplification was detected.

**Table 4 viruses-13-01712-t004:** Comparison of one-tube RT-qPCR kit performances.

	Luna Kit			TaqMan Kit			
	Cq (COBAS)	Cq (N1)	Cq (N2)	Cq (COBAS)	Cq (N1)	Cq (N2)	
Cq	25.3	23.8	24.7	26.9	28.8	29.2	Cq ≤ 30
σ	3.3	5.4	5.6	3.1	3.7	4.2	
n	35	35	35	9	9	9	
Cq	34.8	34.2	35.3	33.5	33.4	34.0	Cq > 30
σ	2.5	2.8	3.2	2.1	2.7	3.0	
n	22	22	22	10	10	10	

Cq, average Cq values measured by the KUK (COBAS) or by our in-house system (N1 and N2); σ, standard deviation; n, number of samples.

**Table 5 viruses-13-01712-t005:** Assay sensitivity.

	High Viral Load Cq ≤ 30		Low Viral Load Cq > 30		Negative	
	COBAS	JKU	COBAS	JKU	COBAS	JKU
positive	53	53	64	36	0	0
inconclusive	0	0	0	13	0	0
negative	0	0	7	22	95	95
Total	53		71		95	

Patient samples taken as naso-pharyngeal swabs assessed for SARS-CoV-2 either by the KUK with the Roche COBAS diagnostic test (COBAS) or our in-house assay (JKU). The 219 samples were divided with a Cq equal or below (53 samples) and above 30 cycles (71 samples) detected either by KUK, JKU, or both. Samples were categorized as negative if no amplification signal was obtained with either platform. Note that samples were categorized as inconclusive if only one of the two probes (N1 or N2) had an amplification product, measured only by JKU.

**Table 6 viruses-13-01712-t006:** Comparison of replicates.

		Replicate 1		Replicate 2		
	Cq (COBAS)	Cq (N1)	Cq (N2)	Cq (N1)	Cq (N2)	
Cq	26.9	28.7	29.4	29.1	29.4	Cq ≤ 30
σ	3.4	4.7	5.1	5.4	4.9	
n	18	18	18	18	18	
Cq	35.0	34.8	36.1	35.1	35.9	Cq > 30
σ	2.4	2.9	3.7	2.9	3.2	
n	49	23	26	25	24	

Shown are the Cq values reported by the KUK (COBAS) or measured by our in-house system (N1 and N2 targets) for replicated measurements of samples collected in either COBAS buffer, NaCl, or PBS and measured on the same day or 1 day apart. Samples were grouped based to the initial Cq value reported by the COBAS system into high viral load (Cq ≤ 30) and low viral load (Cq > 30) samples. Cq, average Cq values; σ, standard deviation of Cq values; n, number of samples. The complete dataset can be found in Appendix A.

**Table 7 viruses-13-01712-t007:** Sample stability.

	High Viral Load Cq ≤ 30			Low Viral Load Cq > 30			Negatives		
	COBAS	JKU		COBAS	JKU		COBAS	JKU	
	Day 1	Day 1	Day 3	Day 1	Day 1	Day 3	Day 1	Day 1	Day 3
positive	12	12	12	8	3	3	0	0	1
inconclusive	0	0	0	0	2	1	0	0	0
negative	0	0	0	0	3	4	15	15	14
Cq	24.0	25.8	26.9	33.2	32.7	33.8	n.a.	n.a.	37.1
σ	4.1	5.1	5.0	2.5	1.1	1.0	n.a.	n.a.	n.a.
CI	2.3	2.9	2.8	1.7	1.2	1.2	n.a.	n.a.	n.a.

Thirty-five patient samples taken in PBS buffer initially measured with the COBAS system (COBAS) were tested by our in-house system (JKU) at day 1 and 3. Shown are the number of samples that were detected as positive, inconclusive, or negative with their respective average Cq values (Cq), standard deviation (σ), and confidence interval (CI).

## Data Availability

The data presented in this study are available in Appendix A.

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
