# Peer review of "Processing Hundreds of SARS-CoV-2 Samples with an In-House PCR-Based Method without Robotics"

_viruses, 2021, doi:10.3390/v13091712_

Round 1

Reviewer 1 Report

The study provided an open system platform and a protocol to perform the PCR testing for SARS-CoV-2 without any robotics. The goal of the protocol was to have high throughput, low cost, and accurate PCR tests without using the expensive commercialized instrument. The study included many aspects, such as the availability of using commercially available testing kits, comparisons on accuracy between the open system platform and the close system of the commercialized instrument, and making a new primer for the internal control. The result showed the open system platform and the proposed protocol can perform accurate PCR tests without any robotics. The results were very accurate when testing high viral load samples (Cq < 30), but became much less accurate on testing low viral load samples (Cq > 30). The new cross exon boundary primer can help to simplify the testing procedures, but some improvement might be needed the Ct values were high and can lead to inconclusive results.

While innovation is not needed, there is no illustration or figure to show the actual use of the plates in the manuscript. Manual operation is not new and is expected to provide great results with a trained technician.  There should have more discussion on the trade-off in the time and labor cost needed to perform the extraction manually versus using a robot.

Other than that, this manuscript provided a lot of imformation that will be beneficial to readers who want to explore non-standard approaches to improve SAR-COV-2 diagnostics or to reduce cost in settings with fewer resources available.

Reviewer 2 Report

The authors present an in-house diagnostic protocol for SARS-CoV-2 detection. The validation of the test was performed correctly and extensively, and the shortcomings in relation to the reference method were proven. Those shortcomings within the in-house method included the impossibility / uncertainty of proving weak positive patients and false negative ones (Cq> 30; Cq 37-45). This should have been critically discussed as well as inconsistency between repeated measurements (Page 4, lines 142-150; page 6, lines 212-215; page 6, lines 188-190). Furthermore, another obvious possible drawback should be addressed, and that is possibility of human error due to fatigue or pain due to extensive pipetting. It is therefore not clear why this test is offered as a good alternative. For these reasons I think that offered protocol has serious limitations, but those limitations have not been discussed. I recommend that you rewrite the article accordingly.

However, the positive and valuable parts of presented protocol are clearly visible:

-The design of the new probe on two distant exon proving successful sampling, extraction, and RT.

-Comparisons of different sample buffers.

-Extensive validations.

Specific comments

Table 1: If RP-2 is negative, doesn’t that mean the test is invalid regardless of the positive results of the virus-specific assays?

Tables 2 and 4: The tables showing the difference between the Cq values obtained with the 2 kits are confusing.

I could not find Suppl. tables.

Round 2

Reviewer 2 Report

My opinion is that the manuscript has been sufficiently improved. Congratulations!